# Indicators of Genotoxicity in Farmers and Laborers of Ecological and Conventional Banana Plantations in Ecuador

**DOI:** 10.3390/ijerph17041435

**Published:** 2020-02-24

**Authors:** Hans-Peter Hutter, Michael Poteser, Kathrin Lemmerer, Peter Wallner, Shifra Shahraki Sanavi, Michael Kundi, Hanns Moshammer, Lisbeth Weitensfelder

**Affiliations:** Department of Environmental Health, Center for Public Health, Medical University, 1090 Vienna, Austria; hans-peter.hutter@meduniwien.ac.at (H.-P.H.); kathrin.lemmerer@meduniwien.ac.at (K.L.); peter.wallner@meduniwien.ac.at (P.W.); shifra.shahrakisanavi@meduniwien.ac.at (S.S.S.); michael.kundi@meduniwien.ac.at (M.K.); hanns.moshammer@meduniwien.ac.at (H.M.); lisbeth.weitensfelder@meduniwien.ac.at (L.W.)

**Keywords:** banana farming, occupational health, pesticides, pesticide sprayers, genotoxicity, cytotoxicity

## Abstract

Banana farming represents an important segment of agricultural production in Ecuador. The health of farmworkers might be compromised by the extensive use of pesticides in plantations applied under poorly regulated conditions. Due to an increased awareness of pesticide-related problems for nature, as well as for worker and consumer health, ecological farming has been established in some plantations of Ecuador. We set out to investigate the occupational health of workers in both conventional and ecological farming. Nuclear anomalies in buccal epithelial cells were used as short-term indicators for genotoxicity and a potentially increased cancer risk in the two groups of farmworkers. By application of the Buccal Micronucleus Cytome Assay (BMCA), we found the frequency of micronuclei in conventional pesticide using farmworkers significantly increased by 2.6-fold, and other nuclear anomalies significantly increased by 24% to 80% (except pyknosis with a non-significant increase of 11%) compared to the farmworkers on ecological plantations. These results demonstrate that ecological farming may provide an alternative to extensive pesticide use with significantly reduced indicators of cancer risk. In conventional farming, improvements in education and instruction regarding the safe handling of pesticides and protective equipment, as well as regulatory measures, are urgently needed.

## 1. Introduction

The farming and exporting of bananas represents a key economic segment of agriculture in many tropical countries. Ecuador is among the top banana producing countries, with a production of more than 6 million tons in 2017 [1]. Within this large agricultural segment, farmers and farmworkers, as well as their relatives in production areas, are often exposed to alarmingly high levels of different pesticides [2,3,4] and carry high risks for acute and chronic health effects; acute symptoms such as poisoning symptoms, skin rashes, etc. are often reported [5]. Chronic health consequences, such as associations with Parkinson’s disease [6] or cancer [7], are also possible (see Alavanja et al. [8] for an overview). Regardless of the manifold possible consequences, workers’ perception of risk differs remarkably from experts’ opinions, especially concerning exposure pathways or health effects [5]. Hence, biomarkers are of great value to identify early biological effects in an exposed population. Specifically for banana farming, a study investigated mutagenicity and cytotoxicity in buccal mucosal cells for farm workers in Brazil [9] and showed a high risk. Similar results have been obtained for coffee plantation workers [10,11,12].

In Ecuador, pesticides are used rather heavily. In 2017 alone, more than 34,250 tons of pesticides were applied for agricultural use [13]. As one of the consequences of this, pesticide contamination in the Guayas river basin, one of Ecuador’s major watersheds, could be shown in 60 % of sampling sites, probably caused mostly by banana and rice farming [14]. Due to economic pressure and lack of information, dangerous pesticides, which have already been abandoned in the agricultural practice of most countries, are still in use [13,14]. The types of pesticides used vary, and the toxicological profiles of the specific chemicals can be very different. For banana farming, the genotoxicity of three common pesticides has been investigated, and evidence has predominantly been provided for DNA breaks by chlorpyrifos and imazalil. A fungizid, thiabendazole, did not cause DNA damage in vitro at any tested concentration [15] but can increase thyroid cancer risk at very high levels.

This report represents the first part of a field study on the acute and chronic health effects observed in small scale banana farmers and farmworkers engaged in conventional (i.e., pesticide applying) and ecological (i.e., pesticide avoiding) agriculture in Ecuador. In this article, we present data on indicators of genotoxicity/mutagenicity and cytotoxicity in ecological and conventional banana farming in Ecuador. Given the results of previous studies [9,10] we anticipated cytotoxic and genotoxic effects of pesticides, but here we provide a detailed overview regarding types of nuclear anomalies. Additionally, we intended to examine whether adherence to specific hygiene measures is connected with less nuclear anomalies.

## 2. Material and Methods

### 2.1. Study Area and Subjects

A questionnaire on pesticide exposure was applied in a total of 71 male farm workers in five locations (Quevedo, La Union, Valencia, La Libertad, Buonavista) of the provinces Los Rios (central) and El Oro (south) in Ecuador. Buccal cells were collected from all participants. Volunteering participants were carefully selected to represent typical pesticide exposure in conventional (i.e., pesticide using) and ecological/organic (i.e., not pesticide using) farming. Thirty-four farmworkers were engaged in conventional farming (pesticide user) and 37 farmworkers worked in ecological/organic farming (non-pesticide user). With a statistical power of 0.8 and a two-sided alpha of 0.05, this sample size is sufficient to detect effect sizes of approximately 2/3 standard deviations.

The mean age on the day of sampling for the two groups was comparable (conventional farmworkers: 44.6 years, SD = 13.6; ecological farmworkers: 44.7; SD = 16.6), but groups differed significantly regarding their educational level, with a higher educational level in ecological farming: almost 40 % of organic farmworkers had a secondary education, while many conventional farmworkers had no education at all (Table 1).

The selection of the study areas and the recruitment of the participants—male small-scale farmers and farmworkers, in the following named “farmworkers”—was carried out with support of several organizations such as Asociación Sindical de Trabajadores Agrícolas Bananeros y Campesinos (ASTAC), the “Federation of Unions of banana workers and farmers”, which in recent years has been acting as a voice for workers in the region Los Ríos, and Unión Regional de Organizaciones Campesinas del Litoral (UROCAL), an umbrella organization of small-scale producers in the southern coastal region of Ecuador (see Figure 1).

### 2.2. Examination Procedure

The study complied with Research Regulations of Ecuador and the Declaration of Helsinki. Each participant gave informed consent (Consent was expressed orally in presence of local ASTAC officials). The study was approved by Comité Ético De Investigación, ASTAC, Quevedo, Ecuador (20-06-017).

Before physical examination, the farmworkers were informed about the methods and the procedure. After registration and assigning an anonymous personal identifier to each volunteer, the weight and size of the study participants were measured. Then, buccal cells were collected as described below.

The questionnaire was designed to obtain information about current and former pesticide use and pesticide exposure, handling of pesticides (incl. hygiene), knowledge and attitudes towards pesticide use and health symptoms experienced in the last 6 months.

### 2.3. Buccal Cells Micronucleus Assay (BMCA)

Several cellular endpoints which reflected genotoxic effects, including relative numbers of micronucleated cells (MN), total number of micronuclei (MNi), nuclear buds and “broken eggs” (BUD), and binucleated cells (BN) as indicators of cytokinetic defects, were blindly evaluated. In addition, we quantified indicators of cytotoxic effects, including cells with condensed chromatin (CC), karyorrhectic cells (KR), karyolytic cells (KL), pyknosis (PY) and basal cells, were counted to assess proliferative activity in the buccal epithelium. According to the standard protocol [16], ≥2000 cells were counted for MN, MNi, and BUD, ≥1000 cells for the other anomalies.

#### 2.3.1. Buccal Cells Sampling and Staining

Buccal cells were collected using spatulas and specimens were transferred on slides, according to the procedure described by Tolbert et al. [17]. Immediately before collecting the cells, participants rinsed their mouths twice with tap water to get rid of possible food residues in the mouth. Buccal mucosa cell samples were collected from both cheeks using separate, moist wooden spatulas. Cell samples were smeared cautiously on the end of each slide with 2–3 drops of distilled water from a sterile pipette. For each subject, slides from both the right and left cheek were prepared and air-dried for 10 min before being stored. After storage in a dry and dark place, the slides were prepared for transport and brought to Vienna for fixation, staining, and evaluation. The slides were placed back to back in Coplin staining jars and fixated using freshly prepared cold methanol and a glacial acetic acid mix (3:1). Subsequently, the samples were stained using the Feulgen Technique as described in [18].

#### 2.3.2. Evaluation of Nuclear Anomalies

The slides were evaluated and scored for nuclear anomalies according to the current standard protocol by Thomas et al. [16], by scorers blinded to the group assignment. Before evaluation, the slides were marked with an ID-code by a person not directly involved in evaluation. From each person, ≥2000 buccal cells were scanned for nuclear aberrations using a 400-fold magnification under both bright-field and fluorescence (using a far red filter) illumination on a Labophot-2 microscope (Nikon, Tokyo, Japan). In the first counting step, frequencies of basal cells, differentiated cells and cells with nuclear anomalies were scored in 1000 buccal cells. In the second step, the counting of differentiated cells was continued for genotoxic nuclear anomalies (MN, total MNi, BUD) until the total count of differentiated cells reached 2000. Photographic images of important nuclear anomalies were taken with a digital microscopic camera (Nikon DS-Fii1, Japan). Concomitantly, all anomalies were cross-checked by another experienced scorer.

#### 2.3.3. Statistical Analysis

Questionnaire-retrieved data were evaluated descriptively. Absolute and percentage frequencies (within the groups of conventional and ecological farmworkers) were calculated for categorical data, whereas mean and standard deviation were calculated for quantitative data.

Buccal mucosa cells were analyzed and assessed by a Generalized Linear Model (GLM) with Poisson counts and a log link. Overdispersion was tested by chi-square tests. The frequencies of nuclear anomalies were considered as dependent variables with the number of counted cells as an offset variable. The primary variable describing pesticide related working conditions was obtained from the question, “working in ecological (non-pesticide-using) farming (yes/no)”. As potential confounders, age (in years), smoking (yes/no), tobacco chewing (yes/no), alcohol consumption (yes/no), and frequency of eating spicy food (five categories from “never” to “daily”) were considered. Participants were asked for possible dental x-rays during the last month as a potential confounder, but none of the farmworkers had received such a treatment. Since nine endpoints were evaluated, *p*-values were Bonferroni-Holm adjusted.

All calculations were done using standard software (IBM SPSS Statistics 26 (IBM, Armonk, NY, USA)). *p* values < 0.05 were considered to be statistically significant.

## 3. Results

Three farmers in the group of conventional farmworkers stated in the questionnaire that they never used pesticides. Hence, they were excluded from further analyses, reducing the exposure group (conventional farmworkers) to 31 men.

Testing all nuclear anomalies in Poisson regressions (*n* = 68 cases), all anomalies except pyknosis were significantly higher in conventional pesticide using farmworkers. No significant differences were found in the frequency of basal cells (Figure 2, Table 2).

Regarding their handling of pesticides, farmers often reported to not apply precautionary measures such as wearing gloves or a mask. Table 3 shows the results of conventional (pesticide-exposed) farmworkers descriptively.

While it was originally planned to compare farmworkers who protect themselves properly with those ones who do not regarding their nuclear anomalies, characteristics of self-protection show that this is not possible; as can be seen in Table 3, hardly any farmworker stated that they use masks or gloves always or more than half of the time. Hence, only the criterion of “changing clothes after spraying”, with 12 farmworkers claiming to do so, can be evaluated statistically. Farmworkers changing clothes after spraying had lower frequencies for all nuclear anomalies except pyknosis. However, statistical significance was only obtained for karyorrhexis and basal cells (Table 4). Correcting for multiple endpoints, these results were no longer statistically significant.

When asked why they do not use a mask, most (75 %) of the non-using farmworkers (*n* = 28) stated that it was not available, 14 % claimed that it was not required, and 11 % stated it was because of being uncomfortable.

## 4. Discussion

The Buccal Micronucleus Cytome Assay (BMCA) is regarded as a suitable test for risk assessment associated with chemical induction of DNA damage [19,20]. A strong correlation between the rate of nuclear anomalies and diseases like cancer and neurodegenerative diseases was demonstrated in several studies [21,22]. Thus, BMCA is a suitable method to detect the genotoxic and short-term-exposure-related-cytotoxic damage in human tissues, which are targeted by potential carcinogens [23].

The present results demonstrate that the group of conventional farmworkers (pesticide users) exhibits significantly higher rates of all nuclear anomalies, except pyknosis. than ecological farmworkers. This corroborates the findings of a recent study about banana farming in Brazil [9].

This fact may be interpreted as an indication that the use of pesticides is linked to a toxicological profile which results in DNA damage as well as cytotoxic effects. Accordingly, cancer risk in exposed farmworkers may rise significantly even in the absence of other signs of acute toxicity.

Our results are in accordance with results obtained in studies investigating genotoxicity in pesticide-exposed populations of the region [24,25,26] as well as increased cancer risk in regions of agricultural use of pesticides [27]. The existing reports also indicate that genotoxic effects and increased cancer risk may not be specific to banana plantations, but can be anticipated in agriculture of various products with a high pesticide application. However, our results clearly show that it is possible to successfully grow tropical fruits under organic conditions with a greatly reduced health risk for farmworkers.

This study once again demonstrates that exposure prevention and individual protection measures have to be applied where possible, which should include educational programs for exposed farm workers and health screenings on a regular basis. Most of the farmworkers did not use masks or gloves, and not even half of the farmworkers changed their clothes after spraying. We have found similar working conditions in our study about coffee plantation workers [11]. Hence, we assume not only a severe lack of knowledge regarding the need of protection measures, but also general neglect of safety measures and regulatory provisions.

The reduced frequency of nuclear anomalies in farmworkers who change clothes after spraying in comparison to those that do not change their clothes indicates that this basic protective measure may already affect the nuclear integrity of buccal cells. Though statistical significance was not reached, if alpha level corrections are applied, all descriptive results point in the direction that not changing clothes does increase the amount of nuclear anomalies. This underlines the importance of meeting safety requirements, including the provision of protective clothing. An observed persistence or worsening of dangerous working conditions is often fostered by a lack of information in the affected population and a lack of general education, as well as a lack of specific training for farmworkers and sometimes also by missing or neglected execution of legal measures [28,29].

The fact that nearly half of the farmworkers return from work to their homes in pesticide-contaminated clothing also leads to the question of to what extent farmworkers could be a risk to family members, such as their wives, who are traditionally responsible for laundry in these communities. Future research is required to investigate possible indirect exposure and health effects in close family members.

The fact that ex-pesticide-users do not have a higher risk for nuclear anomalies than never-users is not surprising. Different types of pesticides than those currently available might have been used in the past, and the renewal rate of the buccal mucosa cells is high [30,31], and hence nuclear anomalies are rather a sign of genotoxic and cytotoxic responses to recent exposure. Consequently, to investigate the effects of long-term exposure for individual cancer risk, different markers seem advisable [32]. Hence, research regarding these longer-lasting markers should be encouraged.

Our study shows that, with respect to short-term indicators of genotoxicity, ecological plantation workers clearly benefit from the abandonment or a reduction of pesticide use. As health problems related to genotoxicity may appear in individuals many years after the exposure to risk factors, communities engaged in conventional farming may be facing a significantly higher socio-economic burden in the treatment of related cancer cases. These important ethical and economical points should be included in considerations when advantages and disadvantages of ecological farming are discussed.

As a limitation to our results, it has to be mentioned that the genotoxic and cytotoxic effects found in this study cannot be assigned to specific pesticide components, which could sometimes have different effects [15], since farmworkers usually are exposed to a mixture of different pesticides. As well as this, nearly half of the farmworkers did not know which pesticides they were exposed to.

Due to the reported difference in the pesticide exposure of males and females [33,34] future studies including women should also be encouraged.

## 5. Conclusions

The results of the BMCA underline the urgent need for protection measures for the affected farmworkers. The impact of pesticide use is not restricted to acute health effects; the results of the BMCA indicate that the occupational exposure to pesticides in banana plantations leads to elevated long-term health risks. The results presented here show that conventional farmworkers exposed to pesticides in banana plantations manifest an increased level of acute genotoxic and cytotoxic damage that can be considered as an indicator for a higher risk of developing cancer. Regarding the handling of pesticides, protection measures are often not used or are unavailable. Educational programs, provision of protective clothing and supervision by authorities, are urgently needed.

## Figures and Tables

**Figure 1 ijerph-17-01435-f001:**
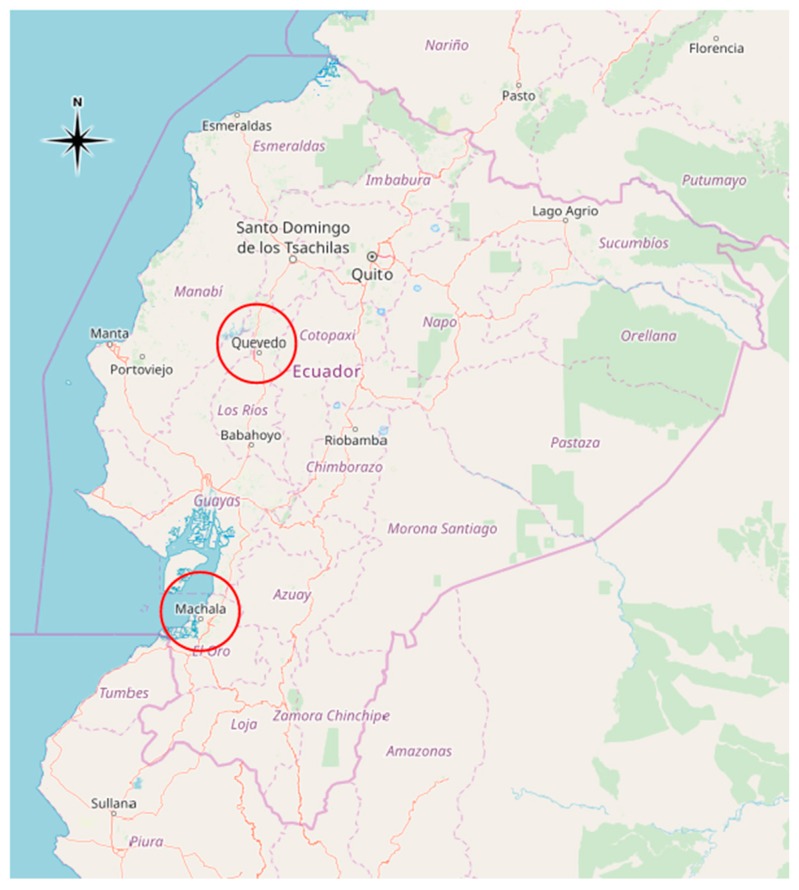
Overview of geographical locations of the study sites in Ecuador.

**Figure 2 ijerph-17-01435-f002:**
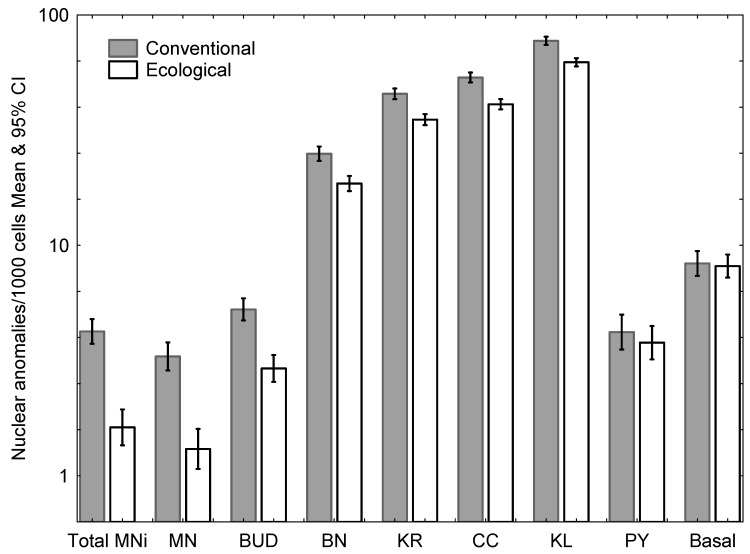
Mean frequency (and 95% confidence intervals) of nuclear anomalies per 1000 cells in the groups of conventional (black) and ecological farmworkers (white). BN: Binucleated cells; KR: Karyorrhexis; CC: Condensed chromatin; KL: Karyolysis; Bottom: PY: Pyknosis; BASAL: Basal cells; Total MNi: Total number of micronuclei; MN: micronucleated cells (number of viewed cells with micronuclei); BUD: Cells with nuclear buds (“broken egg”).

**Table 1 ijerph-17-01435-t001:** Educational level of participants.

Group	No Education	Compulsory Education	Secondary Education
Conventional farming (*n* = 34)	6 (17.6 %)	22 (64.7 %)	6 (17.6 %)
Organic farming (*n* = 37)	1 (2.7 %)	22 (59.5 %)	14 (37.8 %)

**Table 2 ijerph-17-01435-t002:** Crude and adjusted ^1^ means and 95% confidence intervals of nuclear anomalies per 1000 cells in conventional and ecological farmworkers.

Endpoint	Conventional Farmworkers (*n* = 31)		Ecological Farmworkers (*n* = 37)		*p*-Value ^1^
	Adj.Mean (95% CI)	Crude Mean (95% CI)	Adj.Mean (95% CI)	Crude Mean (95% CI)	
Total Mni	4.23 (3.74–4.79)	4.19 (3.71–4.74)	1.62 (1.36–1.94)	1.65 (1.38–1.97)	<0.001
MN	3.30 (2.87–3.79)	3.27 (2.85–3.76)	1.31 (1.07–1.60)	1.34 (1.10–1.63)	<0.001
BUD	5.28 (4.72–5.89)	5.34 (4.79–5.95)	2.92 (2.56–3.35)	2.92 (2.55–3.34)	<0.001
BN	25.00 (23.27–26.86)	25.03 (23.33–26.86)	18.55 (17.20–20.01)	18.54 (17.20–19.98)	<0.001
KR	45.46 (43.11–47.95)	46.48 (44.14–48.95)	35.16 (33.27–37.15)	35.03 (33.17–36.99)	<0.001
CC	53.53 (50.97–56.22)	54.77 (52.23–57.44)	40.97 (38.93–43.12)	40.86 (38.86–42.98)	<0.001
KL	77.26 (74.17–80.47)	76.35 (73.34–79.49)	62.31 (59.78–64.93)	63.30 (60.78–65.91)	<0.001
PY	4.21 (3.53–5.01)	4.19 (3.53–4.98)	3.78 (3.20–4.47)	3.97 (3.38–4.67)	0.389
BASAL	8.35 (7.38–9.46)	8.48 (7.52–9.57)	8.14 (7.25–9.12)	8.08 (7.22–9.05)	0.764

^1^ adjusted for age, smoking (N. conv. = 3, eco. = 3), tobacco chewing (N. conv. = 2, eco. = 0), alcohol consumption (N. conv. = 16, eco. = 14), and spicy food intake (N. conv. = 25, eco. = 12). Total MNi: Total number of micronuclei; MN: micronucleated cells; BUD: Cells with nuclear buds (“broken egg”); BN: Binucleated cells; KR: Karyorrhexis; CC: Condensed chromatin; KL: Karyolysis; PY: Pyknosis; BASAL: Basal cells; Significant (Bonferroni–Holm corrected). Crude and adjusted means differed only slightly, showing that potential confounders corrected for had little impact. Out of the group of ecological farmers (*n* = 37), 14 reported that they had used pesticides in the past. A comparison between “never-users” (ecological farmworkers without pesticide experience in the past) and “ex-users” (ecological farmworkers with pesticide exposure in the past) showed that “ex-users” and “never-users” did not differ significantly in any of the assessed nuclear anomalies.

**Table 3 ijerph-17-01435-t003:** Characteristics of pesticide use and self-protection in conventional farmworkers (*n* = 31).

Exposure Parameter	Unit/Category	Mean (SD) or *n* (%)
Pesticide usage	Years	12.94 (9.47)
Last spraying	Days	36.23 (89.48)
Mixing of pesticides	Never	21 (68 %)
Knowing which pesticides are used	No	14 (45 %)
Mask usage	always or more than 50%	3 (10 %)
Glove usage	always or more than 50%	6 (19 %)
Washing hands at site	always or more than 50%	25 (81 %)
Washing hands immediately at home	always or more than 50%	11 (35 %)
Change clothes after spraying	Yes	12 (39 %)

**Table 4 ijerph-17-01435-t004:** Means ^1^ and 95% confidence intervals of nuclear anomalies in conventional (pesticide using) farmworkers who do and do not change their clothes after spraying.

Endpoint	Change Clothes (No)(*n* = 19)		Change Clothes (yes)(*n* = 12)		*p*-Value
	Mean	95% CI	Mean	95% CI	
Total MNi	4.40	0.35–3.77	3.86	0.41–3.13	0.336
MN	3.55	0.31–2.99	2.83	0.35–2.22	0.148
BUD	5.67	0.39–4.95	4.81	0.46–3.99	0.174
BN	25.65	1.18–23.43	23.74	1.45–21.07	0.321
KR	48.11	1.63–45.02	42.90	1.94–39.25	0.046
CC	55.78	1.75–52.45	52.02	2.14–47.99	0.185
KL	76.53	2.05–72.62	75.54	2.59–70.64	0.770
PY	3.86	0.46–3.06	4.20	0.61–3.16	0.656
BASAL	8.58	0.71–7.29	6.22	0.73–4.95	0.023

^1^ adjusted for age, smoking, tobacco chewing, alcohol consumption, and spicy food intake BN: Binucleated cells; KR: Karyorrhexis; CC: Condensed chromatin; KL: Karyolysis; PY: Pyknosis; BASAL: Basal cells; Total MNi: Total number of micronuclei; MN: micronucleated cells; BUD: nuclear buds (“broken egg”).

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
