# Peer review of "Indicators of Genotoxicity in Farmers and Laborers of Ecological and Conventional Banana Plantations in Ecuador"

_ijerph, 2020, doi:10.3390/ijerph17041435_

Round 1
Reviewer 1 Report
The manuscript of H.P. Hutter et al. entitled “Indicators of genotoxicity in farmers and laborers of ecological and conventional banana plantations in Ecuador” presents interesting and certainly useful subject of study. The manuscript was prepared logically, using correct and elegant English language. The experimental part of the studies was prepared according to the rules applied for such types of studies. I have only one remark, which of course does not diminish the value of presented paper.
Particular objections.
Chapter 2.3.2., line 128: In the sentence “Samples were air dried at room temperature for 10 min and treated with 5 M 37 % (vol/vol) in hydrochloric acid at room temperature for 30 min.” the underlined statement is unclear. I guess, something (some reagent) was omitted in this sentence. Besides, 37% HCl is 12 M.
Author Response
Following the request of another reviewer, we removed the detailed description of Feulgen staining and replaced these sentences by a reference in literature (page 4, line 153).
Reviewer 2 Report
The paper is well-written. The conclusion in particular clarify the fact that the
“study cannot be assigned to specific pesticide components” which is correct. I have a few suggested edits:
Line 73- when talking about the locations it mentions “La Liberdad”- it should be “La Libertad”- I checked on-line.
Line 85- the sentence starts with “Less conventional farmworkers…” suggest keeping the language consistent and use “Ecological farmworkers….” Or “Organic farmworkers…”
Line 92, line 266- when describing ASTAC, Agrícolas needs the accent on the i.
Line 94, line 267 –“Unión Regional des Organizaciones Campesinas del Litoral” it is “Unión Regional de …” no s.
Author Response
Page 2, line 73: now line 84, has been corrected to "La Libertad".
Page 2, Line 85: now page 3, line 98, has "less conventional" has been replaced by "organic farmworkers".
Page 3, line 92 and 266: now line 108 and 354, "Agrícolas" has been written with the accent on the i.
Line 94 and 267: now line 111 and 355, has been corrected to "Unión Regional de Organizaciones..."
Reviewer 3 Report
In this manuscript the authors sought to occupational health of workers in both conventional and ecological farming. Nuclear anomalies in buccal epithelial cells were used as short-term indicators for genotoxicity and potentially increased cancer risk in the two groups of farmworkers. In view of this study, the frequency of micronuclei in conventional pesticide using farmworkers significantly increased 2.6-fold and other nuclear anomalies significantly increased by 24% to 80% compared to farmworkers of ecological plantations. This demonstrate that ecological farming may provide an alternative to extensive pesticide use with significantly reduced indicators of cancer risk. In conventional farming. Overall it is a passably written manuscript that it is an ecological and conventional point of view, given the implication of education and instruction regarding save handling of pesticides and protective equipment. However, there are some points the need further clarification as follows.
Introduction: This part is tedious and hard to follow. These description are occupied too much spaces; these authors should be written more concisely. I suggest the section could be divided to three paragraphs. Material and Methods: Line 86, Table 1and Line 169, Table 1. This part is tedious and disorderly. It is need to be organized again. Results: Line 167-168, no significant differences were found in frequency of basal cells (Table 1, fig.1), however, there are two Table 1 in this manuscript. It is need to be re-written. There are no Table 2 in this part of Results. Discussion: The discussion need more information to show the significant among different studies or references.Author Response
Introduction (now line 36-79): We have shortened the introduction by removing some sentences as well as by transfer of sentences to the discussion section. We were finally able to divide the introduction in three concise parts, as suggested by reviewer 3. The introduction is now thematically structured in study background, pesticide related context and scope of the study.
The detailed description of Feulgen staining, a common standard protocol, has been removed from the Materials and Methods section and was replaced by a reference in literature. We finally combined the description of sampling and staining of buccal cells in a single chapter.
Table numbers have been corrected in the manuscript text as well as in table legends: Page 3, lines 100 and 101; Page 5, line 202; Page 7, lines 211, 226 and 228; Page 8, lines 233, 239 and 245.
The discussion has been slightly extended with the aim to highlight the significance of our results and to show accordance with existing studies: One paragraph (page 9, lines 271-279) and references 24-27 have been added to the text to indicate the supportive findings in existing literature and we emphasized a major outcome of our study (page 9, line 277-279).
To get the discussion thematically structured and clear, we transferred a sentence about working conditions from the introduction to the according section of the discussion.
Reviewer 4 Report
The study investigated the occurrence of genotoxic/mutagenic events in buccal cells of banana farmers working under conventional or ecological (non-pesticide using) conditions. The number of micronucleated cells and several other cellular endpoints which reflect genotoxic effects were addressed in the study. Critical points have been correctly considered in the study design such as the blinding of the evaluator for cell counting and also an adequate number of buccal cells were counted. The most important potential confounders have been addressed and presented results were adjusted for all of them (in total 5). Here, it would help if the results of the potential confounding variables were transparently presented to the reader in a table (how may smokers, tobacco chewers..in the two groups?) Also, giving the unadjusted values for comparison to the adjusted values would improve the data transparency and the reader could see the influence of the single factors on the outcome measurements. A drawback of the study is certainly that no information about the real pesticide exposure is given but the authors are aware of this and mention it in their discussion.
Minor comments: The authors state that the study complied with Research Regulations of Ecuador and the Declaration of Helsinki. Each participant gave informed consent. It is unclear if the participants provided also written informed consent. Also, the study number of the ethical vote should be given in the text.
Author Response
Table 2: We agree that transparency would be enhanced if results for the 5 confounders are shown. However, this would be a large table since results differ of course for each endpoint. Therefore, we follow your suggestion to present in addition to the adjusted means and CI the unadjusted values in Table 2. The number of smokers, tobacco chewers, alcohol consumers and spicy food consumers for each group has been added to the table legend.
A statement about the compliance of our study to the Helsinki Declaration has been added to section 2.2 (page 4, line 115-118) and also a statement about the form how participants expressed their consent has been added as a footnote on page 4 and accordingly indicated in line 116.
We additionally submitted a certificate of compliance to Helsinki ethics standards issued by local authorities to the editorial board.